# Impact of time-dependent data assimilation on ice flow model initialization and projections: a case study of Kjer Glacier, Greenland

**Youngmin Choi**[1,a], **Helene Seroussi**[2], **Mathieu Morlighem**[3], **Nicole-Jeanne Schlegel**[1,b], **and Alex Gardner**[1]

[1]Jet Propulsion Laboratory, California Institute of Technology, Pasadena, CA, USA
[2]Thayer School of Engineering, Dartmouth College, Hanover, NH, USA
[3]Department of Earth Sciences, Dartmouth College, Hanover, NH, USA
[a]now at: Earth System Science Interdisciplinary Center, University of Maryland, College Park, MD, USA
[b]now at: NOAA Geophysical Fluid Dynamics Laboratory, Princeton, NJ, USA

**Correspondence:** Youngmin Choi (yochoi@umd.edu)

**Abstract.** Ice sheet models are often initialized with data assimilation of present-day conditions, in which unknown model parameters are estimated using the inverse method. While assimilation of snapshot observations has been widely used for regional- and large-scale ice sheet simulations, data assimilation based on time-dependent data has recently started to emerge to constrain model parameters while capturing the transient evolution of the system. However, this method has been applied only to a few glaciers with fixed ice front positions, using spatially and temporally limited observations, and has not been applied to marine-terminating glaciers of the Greenland Ice Sheet that have been retreating over the last 30 years. In this study, we assimilate time series of surface velocity into a model of Kjer Glacier in West Greenland to better capture the observed acceleration over the past 3 decades. We compare snapshot and transient inverse methods and investigate the impact of initialization procedures on the parameters inferred, as well as model projections. We find that transient-calibrated simulations better capture past trends and better reproduce changes after the calibration period, even when a short period of observations is used. The results show the feasibility and clear benefits of a time-dependent data assimilation for initializing ice sheet models. This approach is now possible with the development of longer observational records, though it remains computationally challenging.

## 1 Introduction

Mass loss from polar ice sheets has been contributing $\sim 0.7\,\mathrm{mm\,yr^{-1}}$ to global sea-level rise over the past 30 years, and this trend is expected to continue over the next century and beyond (e.g., Mouginot et al., 2019; Rignot et al., 2019; Shepherd et al., 2018; IPCC, 2019). To estimate the future contribution of ice sheets to sea-level rise, accurate ice sheet mass loss projections should be carried out using physics-based numerical models validated against observational data. A lot of progress has been made in ice sheet modeling over the past decades (e.g., Goelzer et al., 2017; Pattyn, 2017) to better capture the present-day state of the ice sheets (Goelzer et al., 2018; Seroussi et al., 2019) and project their future changes. The recent results from the Ice Sheet Model Intercomparison Project for CMIP6 (ISMIP6, Nowicki et al., 2016; Seroussi et al., 2020; Goelzer et al., 2020) demonstrate such improvements but also highlight that many ice sheet model simulations do not capture the mass loss of the Greenland and Antarctic ice sheets observed over the past 30 years (Aschwanden et al., 2021). Although the ISMIP6 experiments provide improved understanding of ice sheet model variability and the different sources of uncertainty, the models' ability to estimate the current state and recent changes of these ice sheets needs to be improved to increase our confidence in the accuracy of these models and to provide more reliable projections.

Typically, model representation of the present-day state of ice sheets and estimation of unknown model parameters have been accomplished using either paleo-climate reconstructions or data assimilation. Paleo-climate reconstructions (e.g., Pollard and DeConto, 2009; Aschwanden et al., 2016) simulate the evolution of ice sheets over long periods, for example, since the last glacier maximum, but often fail at accurately capturing their present-day configuration (Goelzer et al., 2017). Data assimilation (e.g., MacAyeal, 1992; Morlighem et al., 2010), on the other hand, generally consists of time-independent inversions ("snapshot inversion") of model parameters at a given time, without including changes in ice dynamics and ice geometry. It captures the state of an ice sheet at the time of the "snapshot" but does not necessarily reproduce observed temporal trends. In snapshot inversions, a cost function that measures the difference between observed and simulated values is minimized, based on an "adjoint method", which consists of deriving the gradient of the cost function with respect to the unknown, spatially variable parameter. Once the gradient is computed, a standard gradient descend method can be used to minimize the cost function (e.g., MacAyeal, 1993b). While mathematically and computationally challenging, a lot of progress has been made in understanding basal friction and ice shelf rheology using this method (e.g., Larour et al., 2005; Khazendar et al., 2007; Joughin et al., 2009). Snapshot inversions can reproduce a given state of the ice sheet at a single point in time (Morlighem et al., 2013; Gillet-Chaulet et al., 2012), but they can also carry artificial drifts from the initial state of the model, displaying nonphysical artifacts in transient simulations rather than actual trends (Seroussi et al., 2011; Goldberg et al., 2015). To overcome this limitation, modelers have mostly relied on short relaxations (typically a few years), but the required data and periods over which these relaxations should be performed are not clear. Furthermore, relaxing models over a few years or decades can introduce a large deviation from the current ice sheet state (Gillet-Chaulet et al., 2012; Lee et al., 2015).

Alternatively, data assimilation based on automatic differentiation (AD) started to be developed for ice sheet modeling over the past decade to constrain ice flow models over a given period of time using a larger number of observations (Goldberg and Heimbach, 2013; Larour et al., 2014). This method makes it possible to not only estimate the state of the ice at a given point in time, but also to better capture its evolution during the assimilation period by including time-dependent data (Goldberg et al., 2015). AD provides the tools to automatically generate the source code that computes the derivatives of any cost function with respect to any spatially and/or temporally variable model input from a code that solves the direct problem. This allows building time-dependent cost functions and inversion of unknown parameters of transient simulations. This method has been widely applied to ocean and atmospheric circulation models for over 20 years (e.g.,

Wunsch, 2006; Heimbach and Bugnion, 2009), but it is only starting to emerge in ice sheet modeling.

Larour et al. (2014) and Goldberg et al. (2015) introduced the transient data assimilation approach based on AD to invert for poorly constrained variables in ice sheet models. Larour et al. (2014) inferred surface mass balance (SMB) and basal friction of the Northeast Greenland Ice Stream (NEGIS) using surface altimetry data, while Goldberg et al. (2015) calibrated the friction coefficient, boundary stresses, and boundary volume flux based on assimilation of time-varying elevation and velocity of Pope, Smith, and Kohler glaciers in West Antarctica. Although both studies showed promising results in inferring forcings and boundary conditions that yielded the best fit to certain observations, they did not evaluate the models' ability to predict future changes of glaciers. More recently, Goldberg and Holland (2022) used a transient inversion and initialized a coupled ice-sheet–ocean model fitting to velocities and thinning trends and estimated the relative importance of initialization for future projections. Those studies, however, used a relatively short period of about 10 years of observed data with limited spatial coverage, therefore limiting their model domains. In another study, Goldberg et al. (2019) used AD to investigate the impact of spatial patterns of ice shelf melt on Smith Glacier in the Amundsen Sea sector. Morlighem et al. (2021) expanded on this study by generating sensitivity maps of the ice volume above flotation to changes in external forcings and boundary conditions for glaciers in the Amundsen Sea sector. Those studies help determine where changes in different factors would have the largest effect on the mass balance of glaciers and provide guidance on the areas where future geophysical fieldwork should be focused.

The wealth of newly available observations and longer observational periods (Fahnestock et al., 2016; Mouginot et al., 2017; Gardner et al., 2019) now allow for expansion upon previous studies, offering better assessment of transient inversion procedures and guidance on model initialization. Here we present the results of inversions that make use of a time series of observations on Kjer Glacier, West Greenland (Fig. 1). We choose this glacier as the target region for our experiments due to data availability and its recent evolution. The bed topography is relatively well constrained in this region (Morlighem et al., 2017a), and ice front positions and velocity data are available for this glacier from 1985 (Gardner et al., 2019; Wood et al., 2021). After a period of relative stability in the 1980s, Kjer Glacier has been continuously retreating from 1995 and has experienced ∼ 6 km of retreat while more than tripling in velocity over the past decade (Wood et al., 2021). These changes present a well-constrained scenario for investigating the impact of data availability on reproducing recent changes of marine-terminating outlet glaciers. To our knowledge, transient inversions based on AD have not been applied to marine-terminating glaciers of Greenland that require moving boundary capabilities in the ice flow model. The purpose

of this study is to estimate and assess the ability of the ice model initialization process based on the transient inversion to better capture observed changes in a real glacier case, as well as to investigate the impact of initialization procedures and observational data on future projections.

We first describe the observations and the model setup used in the study, along with detailed information on cost functions and the inverse method (Sect. 2). We then present a suite of experiments that investigate the impact of observational data and parameter choices in the initialization procedure on the model's ability to reproduce these observations and short-term projections (Sect. 3). We continue with the discussion (Sect. 4) and conclusions (Sect. 5) on the implications of our method towards hindcasts and projections of ice sheet evolution.

## 2  Data and model

### 2.1  Observations

Our model domain covers the catchment basin of Kjer Glacier. The main branch of Kjer Glacier retreated continuously from 1985 to 2007 (Wood et al., 2021) with a fairly slow and regular increase in velocity during that period (Gardner et al., 2019). After 2007, the retreat accelerated and the velocity of the centerline increased by more than 3 times. In response to this acceleration, the glacier thinned ∼ 50 m over an 11-year period. To initialize the model of Kjer Glacier in 2007, we use surface and bed elevation from BedMachine v3 (Morlighem et al., 2017a). For models that start in different years, we use surface elevation data interpolated from various datasets including an aerial photography digital elevation model (Korsgaard et al., 2016) and elevation change derived from satellite radar altimetry (Sørensen et al., 2018). We use the geothermal heat flux from Greve (2019) and surface temperature from RACMO2.3p2 (Noël et al., 2018) to calculate the steady-state glacier temperature used for the initial state in our experiments. We then force the model using the surface mass balance from RACMO2.3p2 (Noël et al., 2018) and observed ice front positions (Wood et al., 2021). In experiments where calving is simulated instead of being prescribed (described below), undercutting rates derived from thermal forcing are from Wood et al. (2021).

We use two observational datasets for the cost functions defined in Sect. 2.4: ice velocities and ice front positions. Surface velocities are from the ITS_LIVE project (Gardner et al., 2019); the annual mean surface velocities between 1985 and 2018 are derived from Landsat imagery. Spatial coverage varies each year but is nearly complete for the years following the launch of Landsat 8 in 2013. The annual ice front positions from Wood et al. (2021) are used to either force the ice front positions or to measure the misfit of modeled and observed ice front positions in the cost function.

### 2.2  Model setup

We use the Ice-sheet and Sea-level System Model (ISSM, Larour et al., 2012) to simulate the evolution of Kjer Glacier based on the shelfy-stream approximation (SSA, MacAyeal, 1989), a simplification of the Stokes equations describing the stress balance of an ice sheet, which greatly reduces computational expense while being valid for fast-flowing areas, such as outlet glaciers of Greenland. We use the SSA due to the high memory requirement of the CoDiPack library (Sagebaum et al., 2019) used in ISSM for AD computation (Morlighem et al., 2021). The mesh resolution of our domain varies from 500 m near the coast to 20 km inland, resulting in ∼ 8000 elements.

The basal shear stress, $\tau_b$, is described by a modified regularized Coulomb friction law, as it has been shown to better reproduce observed acceleration of glaciers, including Kjer Glacier, compared to other friction laws (Joughin et al., 2019; Choi et al., 2022). The equation describing the regularized Coulomb friction law is

$$\tau_b = C N \frac{|u_b|^{\frac{1}{m}-1} u_b}{(|u_b| + k N^m)^{\frac{1}{m}}}, \tag{1}$$

where $C$ is a friction parameter that we invert for in this study, $N$ the effective pressure, $u_b$ the ice basal velocity, and $m$ the velocity exponent that is set to 3 in this study. We assume that $N$ is equal to the ice pressure above hydrostatic equilibrium and define $k$ such that $k N^m$ is equal to 300 m yr$^{-1}$, corresponding to a velocity threshold used to mark the transition between Weertman and Coulomb friction regimes (Joughin et al., 2019).

The ice viscosity is defined using Glen's law (Glen, 1955):

$$\mu = \frac{B}{2\dot{\varepsilon}_e^{1-\frac{1}{n}}}, \tag{2}$$

where $B$ is the ice viscosity parameter, $\dot{\varepsilon}_e$ the effective strain rate, and $n$ Glen's law exponent set equal to 3.

To simulate the evolution of calving fronts, we use the von Mises tensile stress calving law (Morlighem et al., 2016) to calculate the calving rate at each time step (Choi et al., 2018). The calving rate, $c$, is assumed to be proportional to the tensile von Mises stress, $\tilde{\sigma}$, which accounts only for the tensile component of the stress in the horizontal plane,

$$c = \|\boldsymbol{v}\| \frac{\tilde{\sigma}}{\sigma_{\max}}, \tag{3}$$

with

$$\tilde{\sigma} = \sqrt{3} \, B \, \tilde{\dot{\varepsilon}}_e^{1/n}, \tag{4}$$

where $\sigma_{\max}$ is a stress threshold that is calibrated to fit observations and $\tilde{\dot{\varepsilon}}_e$ is the effective tensile strain rate as described in previous studies (Morlighem et al., 2016; Choi et al., 2018).

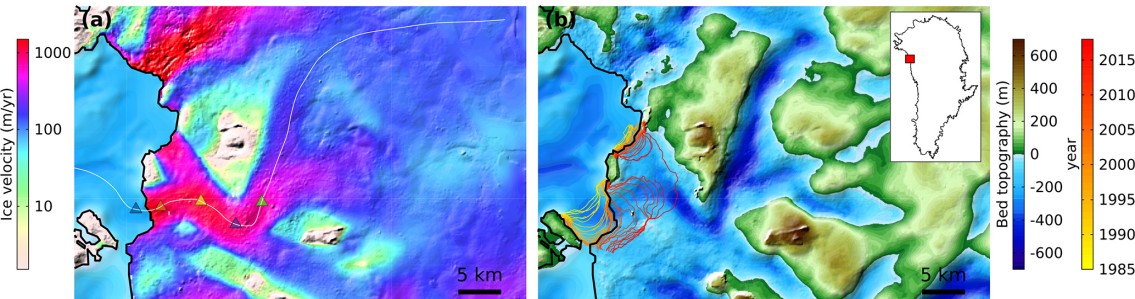

**Figure 1. (a)** Surface velocity and ice front position (black line) of Kjer Glacier, northwest Greenland, in 2007 (Gardner et al., 2019). The white line shows the centerline of Kjer Glacier, and colored triangles denote the locations of points used for velocity comparison in Figs. 3, 5, and 7. **(b)** Bedrock topography (Morlighem et al., 2019) of the same region with observed ice front positions (Wood et al., 2021) from 1985 to 2018. The black line shows the ice front position in 2007. The inset shows the location of Kjer Glacier.

## 2.3 Inverse method

We use the term "snapshot inversion" to refer to the inverse method using a single-time observation and "transient inversion" when using observational data at multiple times in a time-evolving simulation. To invert for unknown parameters (e.g., friction coefficient and ice viscosity parameter) in the model, we minimize cost functions that capture the misfit between modeled and observed fields by computing the gradient of the cost function with respect to corresponding control parameters and applying a gradient descent method (Morlighem et al., 2010). For the snapshot inversions, the gradient is calculated analytically and used in the adjoint method. This approach has been widely applied in glaciology as the stress balance equations of ice flow are considered self-adjoint when the dependence of ice viscosity on strain rates is ignored (e.g., Morlighem et al., 2013). For the transient inversions, however, gradients cannot be calculated analytically; they require a time-dependent adjoint model, which can be done with AD (Griewank and Walther, 2008). ISSM uses the overloaded operator framework of the CoDiPack library (Sagebaum et al., 2019), in which the operations are recorded in memory and a single reverse sweep is performed with the chain rule to compute the gradient of the cost function with respect to poorly constrained model inputs.

Here, using a transient inversion, we invert for the friction coefficient and ice viscosity parameter over the entire model domain, similar to what is commonly done in ice sheet modeling (MacAyeal, 1993a; Rommelaere and MacAyeal, 1997). These two quantities are spatially variable and are either assumed to be constant in time or temporally variable depending on the experiment as described below. We also infer the calving parameter $\sigma_{\max}$ for experiments simulating evolving calving front positions: this parameter is spatially uniform for each basin and assumed to be constant in time. This approach allows us to better understand the physical processes (e.g., changes in friction coefficient or viscosity parameter) involved in reproducing the state of the ice stream at a given

time and the time-evolving state that accounts for the transient nature of observations.

## 2.4 Cost function

We use several cost functions to quantify the misfit between model and observations. For snapshot inversions, we define a cost function that measures the misfit between observed ($\boldsymbol{v}_{\mathrm{obs}}$) and modeled velocity ($\boldsymbol{v}$) for a given time as

$$\mathcal{J}_s(C, B) = \gamma_1 \int_{\Gamma_s} \frac{1}{2} \|\boldsymbol{v} - \boldsymbol{v}_{\mathrm{obs}}\|^2 \, \mathrm{d}\Gamma_s + \gamma_2 \int_{\Gamma_s} \frac{1}{2} \ln\left(\frac{\|\boldsymbol{v}\| + \varepsilon}{\|\boldsymbol{v}_{\mathrm{obs}}\| + \varepsilon}\right)^2 \, \mathrm{d}\Gamma_s$$
$$+ \gamma_3 \int_{\Gamma_b} \frac{1}{2} \|\nabla C\|^2 \, \mathrm{d}\Gamma_b + \gamma_4 \int_{\Gamma_b} \frac{1}{2} \|\nabla B\|^2 \, \mathrm{d}\Gamma_b, \tag{5}$$

where $\varepsilon$ is a minimum velocity to avoid singularities and $\Gamma_s$ is the ice surface. The first term is the mean square error, the second term quantifies the difference between the observed and modeled velocity on a logarithmic scale, and the last two terms are regularizing terms that penalize large gradients in the inferred parameter to avoid overfitting. $\gamma_i$ (where $i = 1\ldots4$) represents weight parameters calibrated by L-curve analyses (Hansen, 2001), as described in Appendix A.

For transient inversions, we define a cost function that quantifies the spatiotemporal misfit between the model and observations. The cost function includes time series of surface velocities as `TS1`

$$\mathcal{J}_{t1}(C, B) = \gamma_1 \sum_{t_i} \int_{\Gamma_s} \frac{1}{2} \|\boldsymbol{v}(t) - \boldsymbol{v}(t)_{\mathrm{obs}}\|^2 \, \mathrm{d}t \, \mathrm{d}\Gamma_s$$
$$+ \gamma_2 \sum_{t_i} \int_{\Gamma_s} \frac{1}{2} \ln\left(\frac{\|\boldsymbol{v}(t)\| + \varepsilon}{\|\boldsymbol{v}(t)_{\mathrm{obs}}\| + \varepsilon}\right)^2 \, \mathrm{d}t \, \mathrm{d}\Gamma_s$$
$$+ \gamma_3 \sum_{t_i} \int_{\Gamma_b} \frac{1}{2} \|\nabla C\|^2 \, \mathrm{d}\Gamma_b + \gamma_4 \sum_{t_i} \int_{\Gamma_b} \frac{1}{2} \|\nabla B\|^2 \, \mathrm{d}\Gamma_b, \tag{6}$$

where $t_i$ represents the years for which we have partial or complete velocity coverage. We use this cost function to si-

multaneously invert for the friction coefficient ($C$) and ice viscosity parameter ($B$) that best fit the observations.

Additionally, we define a cost function that measures the misfit between the observed and modeled ice front positions, represented by the level-set field ($l$) in ISSM (Bondzio et al., 2016) as TS2

$$\mathcal{J}_{t2}(\sigma_{\max}) = \gamma_5 \sum_{t_i} \int_{\Gamma_s} \frac{1}{2} \|l(t) - l(t)_{\text{obs}}\|^2 \, dt \, d\Gamma_s, \tag{7}$$

where $\gamma_5$ is the weight parameter. We use this cost function to invert for the stress threshold ($\sigma_{\max}$) in the calving law.

## 3 Experiments and results

Here we present a series of experiments to assess the model initialization capabilities using transient inversion and the role of the initialization procedure and observational data (Table 1). We first compare initialization methods, namely snapshot and transient inversions, and their ability to accurately predict the acceleration and mass loss of Kjer Glacier. Next, we explore the impact of the length of the initialization period and the time dependence of control parameters on model initialization and projection results. For model initialization and projection, we force the position of the ice front with observations to reduce the impact of errors introduced by the calving law in the modeled ice front positions. Finally, we investigate the possibility of expanding our initialization capabilities to include the stress threshold of the calving law as an additional control parameter by simulating the front migration in the model. Experiment details are summarized in Table 1.

### 3.1 Comparison of snapshot and transient inversions

We first compare the models' predictive skills for simulations initialized with snapshot and transient inversions. Projections start in 2007, the year in which the Greenland Ice Mapping Project (GIMP, Howat et al., 2014) provides a complete spatial mapping of surface elevation. We initialize two simulations using either snapshot inversion in 2007 (experiment SI) or transient inversion over the 1985–2007 period (experiment TI).

For the snapshot inversion, we invert for the friction coefficient ($C$ in Eq. 1) and the ice viscosity parameter ($B$ in Eq. 2) using the cost function $\mathcal{J}_s$ (Eq. 5) with 2007 velocities. We use an initial value of $B$ estimated from the modeled temperature, using the temperature-dependent relationship table from Cuffey and Paterson (2010). Ice temperature is calculated based on the enthalpy formulation (Aschwanden et al., 2012; Seroussi et al., 2013), using geothermal heat flux from Greve (2019) and surface temperature from RACMO2 (Noël et al., 2018). After the snapshot inversion, $B$ decreases by up to 10 % along the margins compared to the value estimated

with the thermal model, while limited changes are found in other areas (not shown here).

For the transient inversion, starting with the inferred friction coefficient from the snapshot inversion and the temperature-based viscosity parameter as initial values, we simultaneously optimize the friction coefficient and viscosity parameter by minimizing the cost function $\mathcal{J}_{t1}$ (Eq. 6). We use the observed velocities from 1985 to 2007 in this cost function, forcing the model with surface mass balance and observed ice front positions for the same period. For this experiment, time-independent control parameters are used in the optimization.

Once these inversions are performed, we run the models from 2007 to 2018 (i.e., beyond the hindcast period used to constrain the models), forcing them with surface mass balance from RACMO2.3p2 (Noël et al., 2018) and observed ice front positions (Wood et al., 2021). This approach limits uncertainties introduced by the calving parameterization and eliminates the need to reconcile surface mass balance with the mass loss estimated by the calving law. As a result, we can focus on the role of initial conditions inferred from the inverse methods.

Both snapshot and transient inversions capture the observed 2007 velocity with good accuracy (Fig. 2). The root mean square error (RMSE) with observed velocities of 2007 for the catchment basin is 51 m yr$^{-1}$ for the SI, which is slightly lower than the RMSE from the TI (57 m yr$^{-1}$). The misfit to observed velocity is concentrated near the ice front and along the shear margins of both branches of Kjer Glacier. The transient inversion reduces this misfit along the shear margins, although the overall misfit for the northern branches increases.

Figure 3 shows that the observed annual velocity of Kjer Glacier more than triples from 2007 to 2018 along the ice stream. This observed acceleration is not well captured by the SI (Fig. 3a): the modeled velocities are consistently slower than the observed velocities, with a range of 86 %–103 % of observed velocities in 2007 decreasing to 73 %–77 % in 2016, indicating a failure to capture the observed acceleration. TI shows a better agreement with observed changes in velocity (Fig. 3b): the modeled-to-observed-velocity ratio in 2007 is 105 %–113 %, and it is between 92 % and 105 % in 2016.

The modeled changes in grounded ice mass during the 2007–2018 period are compared against the estimated changes in mass for Kjer Glacier from Mouginot et al. (2019) (Fig. 4). The snapshot-calibrated simulation underestimates the observed mass loss over the past 11 years, likely due to the underestimation of the ice velocity. The model loses only 34 Gt of ice, while the observed mass loss from 2007 to 2018 is $51 \pm 3.6$ Gt. When the model is transiently calibrated with velocity data from 1985 to 2007, the changes in ice mass are well captured and remain within the error margin of observation during this period. After 2007, the modeled mass loss remains in good agreement with observed mass loss, al-

**Table 1.** List of experiments, control variables, observations, and calibration periods. Controls that vary in time are shown as functions of $t$ (e.g., $C$ is a static friction coefficient, whereas $C(t)$ is temporally variable).

| Experiment name | Control variables | Cost function | Calibration period |
|---|---|---|---|
| SI (snapshot inversion) | $C$, $B$ | $\mathcal{J}_s$ | 2007 |
| TI (transient inversion) | $C$, $B$ | $\mathcal{J}_{t1}$ | 1985–2007 |
| TI_PD1 | $C$, $B$ | $\mathcal{J}_{t1}$ | 1992–2007 |
| TI_PD2 | $C$, $B$ | $\mathcal{J}_{t1}$ | 1997–2007 |
| TI_PD3 | $C$, $B$ | $\mathcal{J}_{t1}$ | 2002–2007 |
| TI_PD4 | $C$, $B$ | $\mathcal{J}_{t1}$ | 2010–2013 |
| TI_CTR1 | $C$, $B$ | $\mathcal{J}_{t1}$ | 1985–2018 |
| TI_CTR2 | $C(t)$, $B$ | $\mathcal{J}_{t1}$ | 1985–2018 |
| TI_CTR3 | $C$, $B(t)$ | $\mathcal{J}_{t1}$ | 1985–2018 |
| TI_CTR4 | $C(t)$, $B(t)$ | $\mathcal{J}_{t1}$ | 1985–2018 |
| TI_Calving | $\sigma_{\max}$ | $\mathcal{J}_{t2}$ | 1985–2018 |

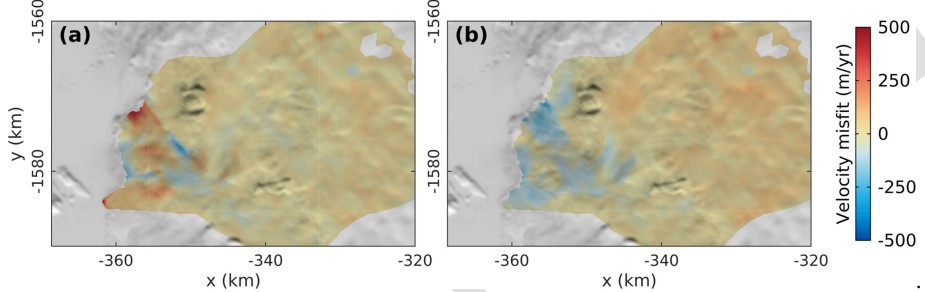

**Figure 2.** Difference between observed and modeled velocities (observation minus model) in 2007. **(a)** Snapshot inversion (SI) and **(b)** transient inversion (TI) calibrated with velocity data. Observed velocities are from Gardner et al. (2019)

though it is no longer within the error margin of observation during 2013–2015.

## 3.2 Impact of the observational period used in the transient inversion

Despite the significant increase in the amount of remote sensing observations over the past decade, there still remain large spatial and temporal gaps over the Greenland Ice Sheet (Gardner et al., 2019), especially prior to the launch of Landsat 8 in 2013. For glaciers with shorter observational records or when computational resources are limited to run AD over long periods, the calibration period needed to initialize simulations should be investigated to determine the benefits of the transient inversion approach. Using a relatively complete record of observations since 1985 for Kjer Glacier, we investigate the impact of the observational period used in the transient inversion on near-future projections. We compare the previous initialization, performed using 23 annual maps of surface velocity observations from 1985 to 2007, with transient inversions using the same velocity datasets but over reduced periods – (1) 1992–2007 (TI_PD1), (2) 1997–2007 (TI_PD2), and (3) 2002–2007 (TI_PD3), respectively – to calibrate our control parameters. After the inversions, we

run three additional simulations from 2007 to 2018 using the same external forcings (SMB and ice fronts) used in SI and TI and evaluate their performance in terms of matching observations after 2007 (i.e., observations that were not "seen" by the model during the calibration period). In addition, we initialize the model using velocities from 2010 to 2013 (TI_PD4) to explore the impact of including more variability in the observations during the calibration period.

Figure 5 shows the modeled velocities with different calibration periods for transient inversions. The modeled velocities from 2007 to 2018 are in better agreement with observations for the three transient-calibrated simulations calibrated before 2007 (TI_PD1–TI_PD3), relative to the snapshot-calibration run (SI). The ratios of modeled to observed velocities in 2016 are similar between the three simulations, ranging between 89 % and 105 %. When only 5 years of observations are used in the cost function, the model overestimates the acceleration closest to the calving front (red in Fig. 5c), where ice velocities are sensitive to the calving front retreat for 2007–2010. This overestimated acceleration leads to a larger mass loss in the future simulation than in the others cases (Fig. 6). However, the modeled mass losses for all three simulations are still better captured, compared to the

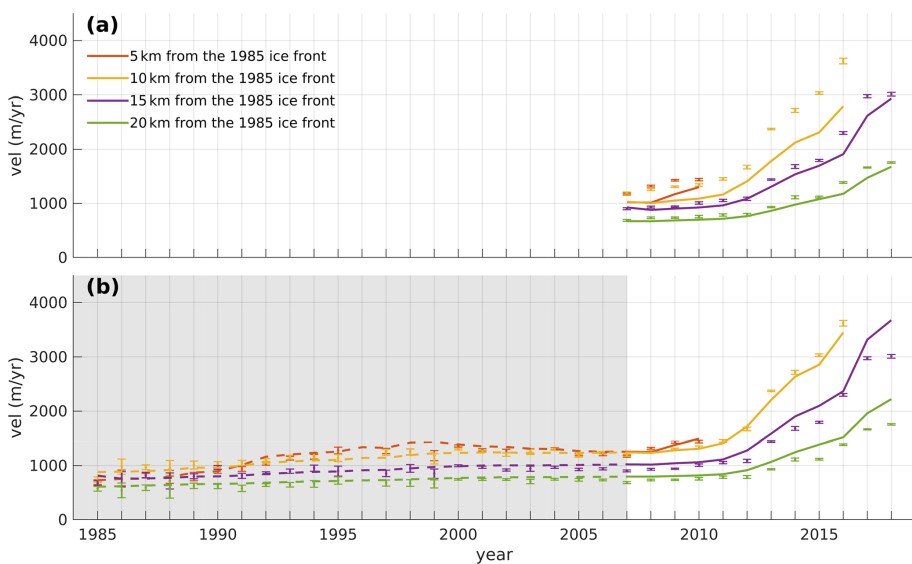

**Figure 3.** Comparison of observed (dots with error bars, Gardner et al. (2019)) and modeled (solid and dashed lines) velocities for experiments with **(a)** snapshot inversion (SI) and **(b)** transient inversion (TI) using a time series of velocity. Colors correspond to the triangle locations shown in Fig. 1. The ice front retreated inland of the red and yellow observation locations in 2010 and 2016, respectively. The gray box indicates the period of model calibration for TI.

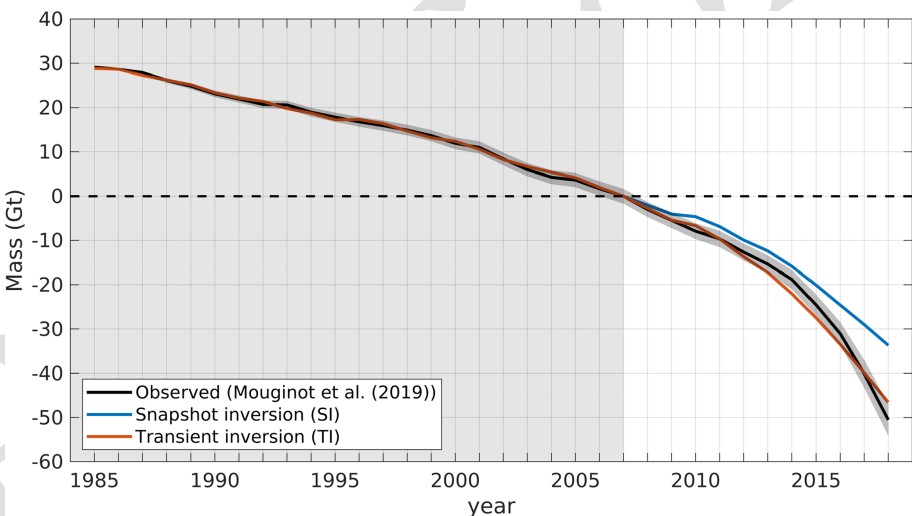

**Figure 4.** Observed and modeled mass change of Kjer Glacier relative to 2007. The black line with a gray error envelope shows the observed mass changes and their uncertainties. The blue and red lines show modeled mass changes from SI and TI, respectively. The gray box shows the period of model calibration for TI.

snapshot-calibration run, regardless of the observational period used for the cost function. When using only 3 years of observations during the rapid acceleration of 2010–2013 for the transient inversion (TI_PD4), the model still effectively captures the acceleration after the inversion period but displays more variability and increased acceleration.

## 3.3 Time-dependent control parameters

Our transient inversion framework allows for the estimation of control parameters that vary not only in space, but also in time (Larour et al., 2014; Goldberg et al., 2015). In all previous calibration experiments, the control parameters were held constant in time. We now investigate the impact of using time-dependent control parameters. To do this, we run three additional simulations with temporally varying control parameters: (1) the viscosity parameter is time-invariant while

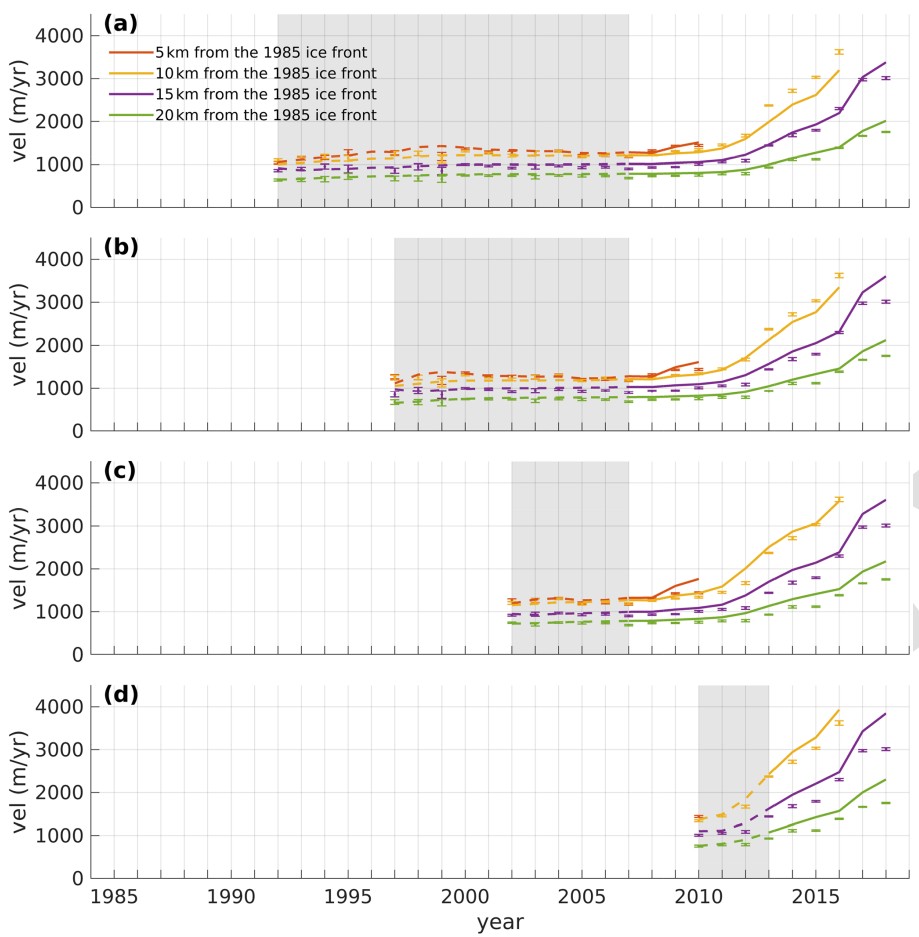

**Figure 5.** Same as Fig. 3 but with transient inversions using velocity data for **(a)** 1992–2007 (TI_PD1), **(b)** 1997–2007 (TI_PD2), **(c)** 2002–2007 (TI_PD3), and **(d)** 2010–2013 (TI_PD4).

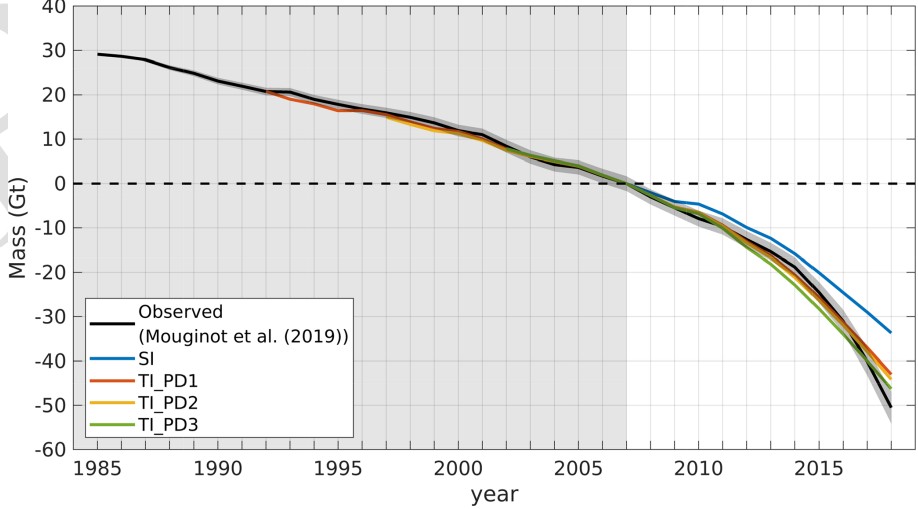

**Figure 6.** Same as Fig. 4 but based on transient inversions using velocity data for 1992–2007 (red, TI_PD1), 1997–2007 (yellow, TI_PD2), and 2002–2007 (green, TI_PD3). The blue line shows modeled mass change from SI shown in Fig. 4. The gray box indicates the period of model calibration for the transient inversion (TI).

the friction coefficient evolves every year (TI_CTR2); (2) the friction parameter is assumed time-invariant, but the viscosity parameter $B$ is allowed to vary every year (TI_CTR3); and (3) both parameters are allowed to vary every year (TI_CTR4). Time-dependent parameters have constant values during each year (i.e., they are allowed to change at the beginning of each year). For these simulations, we use all available annual velocity observations, from 1985 to 2018, to investigate the time dependence of control parameters during both the stable and acceleration periods of Kjer Glacier.

Since the models in these simulations are calibrated with observed velocities from 1985 to 2018, the modeled velocities during this period are all in a good agreement with observations (Fig. 7). To compare results between different simulations in more detail, we calculate the RMSE with observed annual velocities (Fig. 8). Although the RMSE values in 2017 and 2018 are relatively higher than other years in all simulations, those values significantly decrease when both parameters are allowed to vary in time. For other years, reductions in RMSE are relatively small compared to the one for the 2018 RMSE value.

We compare the inverted viscosity parameter fields between simulations. Figure 9a shows the depth-averaged viscosity parameter calculated from the thermal model, which was used as the initial value of $B$ for snapshot and transient inversions. We display changes in viscosity parameter at the end of each experiment, in 2018, compared to the initial viscosity parameter. The initial viscosity parameter indicates softer ice in the shear margins compared to the center of the ice stream. In the transient inversion-based simulations, the viscosity parameter in the shear margins decreases by up to 45 % to match observed accelerations (Fig. 9b–d). The decrease in viscosity parameter also occurs upstream, but the reduction is relatively small. The spatial patterns of the reduction in the viscosity parameter are similar between transient inversions, regardless of the time dependency of parameter, indicating a similar impact of changes in rheology on ice velocity.

The initial basal stress in 1985 and changes in basal stress during the calibration period are shown in Figure 10. The spatial patterns of initial basal stress for TI_CTR1 and TI_CTR3 are smoother than those for TI_CTR2 and TI_CTR4. When both control parameters are static in time (TI_CTR1), the basal stress starts to decrease near the margin and keeps decreasing along $\sim 15$ km of the ice stream by up to 50 % until 2018. This pattern of changes in basal stress is similar to the result with the static friction coefficient and the varying viscosity parameter (Fig. 10) and is caused by changes in the effective pressure and ice velocity that impact the basal stress (see Eq. 1). For simulations with a transient friction coefficient, however, the basal stress field shows extremely large and unrealistic changes of more than 100 % over periods of just a few years during the calibration period, suggesting that it is overfitting the observations.

## 3.4 Optimization of the calving law parameter

Calving is a critical process that controls the dynamics of marine-terminating glaciers (e.g., Rignot et al., 2013; Bondzio et al., 2017; Benn et al., 2007). In ice sheet models, calving is generally represented by a calving law that includes one or more parameters that need to be calibrated (Choi et al., 2018). Calibrating these parameters is essential, not only to reproduce observed changes of glaciers, but also to project future ice sheet changes with moving boundaries. We investigate here whether the stress parameter ($\sigma_{max}$) in a von Mises stress calving law (Morlighem et al., 2016) can be inferred from the observations as part of the model initialization framework. Instead of forcing the ice front position to follow observed terminus migration, we now let a calving law determine a calving rate that dynamically changes the model boundary. To run this experiment, we use here the model calibrated with the static friction coefficient and viscosity parameter from 1985 to 2018, and we use the cost function $\mathcal{J}_{t2}$ (Eq. 7) to calibrate the spatially and temporally constant stress parameter, $\sigma_{max}$, for 1985–2018. All other parameters remain unchanged during the inversion.

Observations show that the main branch of Kjer Glacier retreated continuously from 1985 to 2007 and slowed down at the location of a small ridge in the underlying bedrock. After 2007, it retreated past the ridge and continued retreating again until 2018. The northern branch of Kjer Glacier was stable at the ridge until 2016, but the two branches merged into a single ice front around 2017 when the main branch retreated towards the northern branch. The calibrated model reproduces the observed retreat of the main branch relatively well (Fig. 11), although it could not reproduce the retreat of the smaller northern branch, similar to the recent modeling study of Choi et al. (2021). The calibrated value of the stress threshold for the main branch, $\sigma_{max} = 320$ kPa, is similar to the value $\sigma_{max} = 306$ kPa, determined with manual calibration for the 2007–2018 period in Choi et al. (2021).

## 4 Discussion

In this study, we explore the benefits of transient inversions to initialize ice flow simulations as well as the impact of the observation period used and choices made for control parameters. The snapshot inversion reproduces the observed 2007 velocity slightly more accurately than the transient inversion because the snapshot inversion optimizes the model to the 2007 velocity only, while the transient inversion calibrates the model to observational data over multiple years. Our results show that the model calibrated with the transient inversion better reproduces the time-varying behavior of the glacier during the initialization, which improves confidence in the model's ability to provide realistic near-future projections, although the calibration error and its influence on the model projections still need to be quantified. Future research

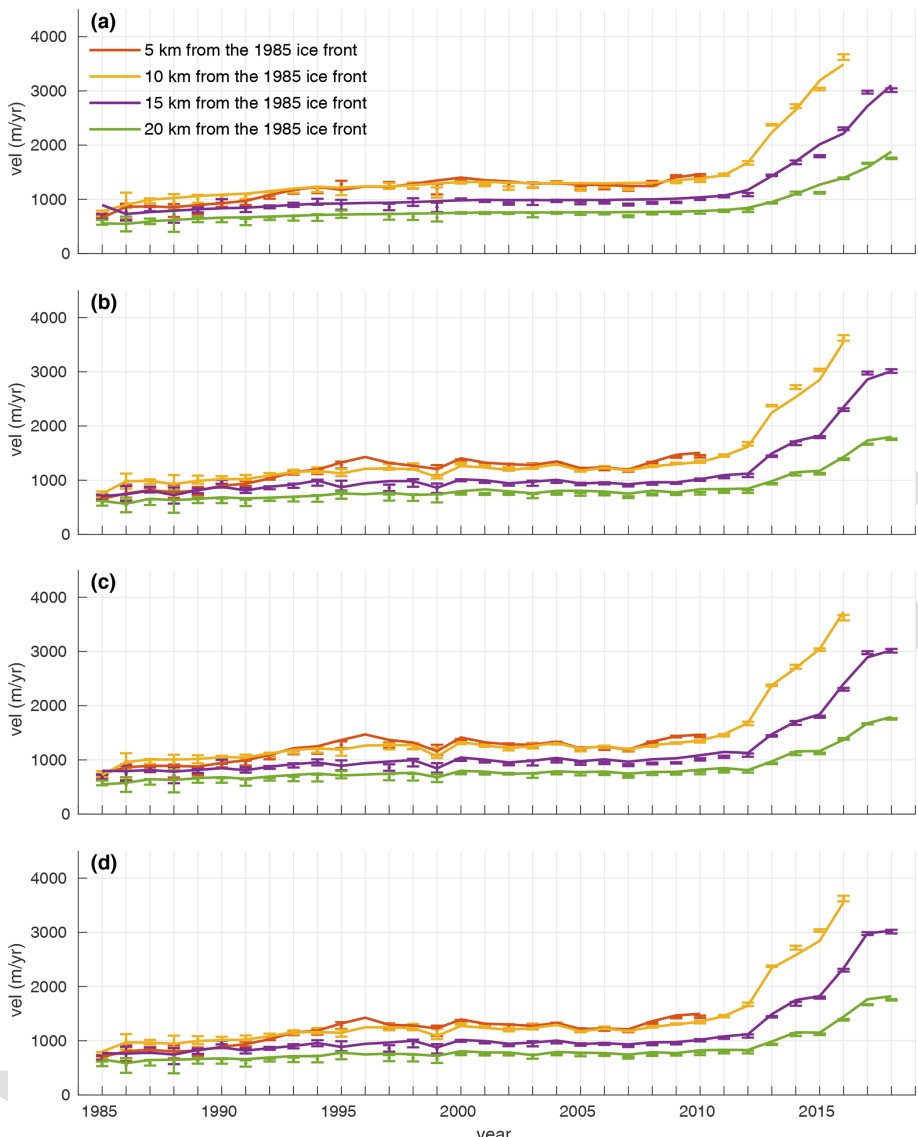

**Figure 7.** Same as Fig. 3 but with transient inversions from 1985 to 2018 calibrating time-variant parameters: **(a)** static friction coefficient and viscosity parameter (TI_CTR1), **(b)** transient friction coefficient and static viscosity parameter (TI_CTR2), **(c)** static friction coefficient and transient viscosity parameter (TI_CTR3), and **(d)** transient friction coefficient and viscosity parameter (TI_CTR4).

should focus on quantifying calibration uncertainties in inversions by sampling the entire variability within the parameter space and evaluating the impact of parameter uncertainty propagation on projections.

To validate our approach and determine its potential applicability to other glaciers, we conducted additional simulations for Sverdrup Glacier, which is located near Kjer Glacier but has exhibited a different behavior over the past 30 years. Unlike Kjer Glacier, Sverdrup Glacier was stable until 2000 and has since experienced a relatively steady acceleration for about 20 years (Fig. 12). When the model is calibrated using the snapshot inversion, the observed acceleration of Sverdrup Glacier's centerline from 2007 to 2018

is overestimated, with modeled velocities being over 20 % faster than the observations beginning in 2011 (red and yellow in Fig. 12a). However, the transient-calibrated simulation, which used velocities from 1985 to 2007, better replicates the observed changes after 2007, with the modeled velocities remaining within 10 % of the observed velocities until 2016. While this simulation also overestimates the velocities for 2017–2018, other factors (e.g., ice front retreat) may have contributed to changes in velocity beyond friction and rheology. Overall, these results for Sverdrup Glacier are consistent with those for Kjer Glacier, demonstrating that the transient-calibrated simulations better reproduce obser-

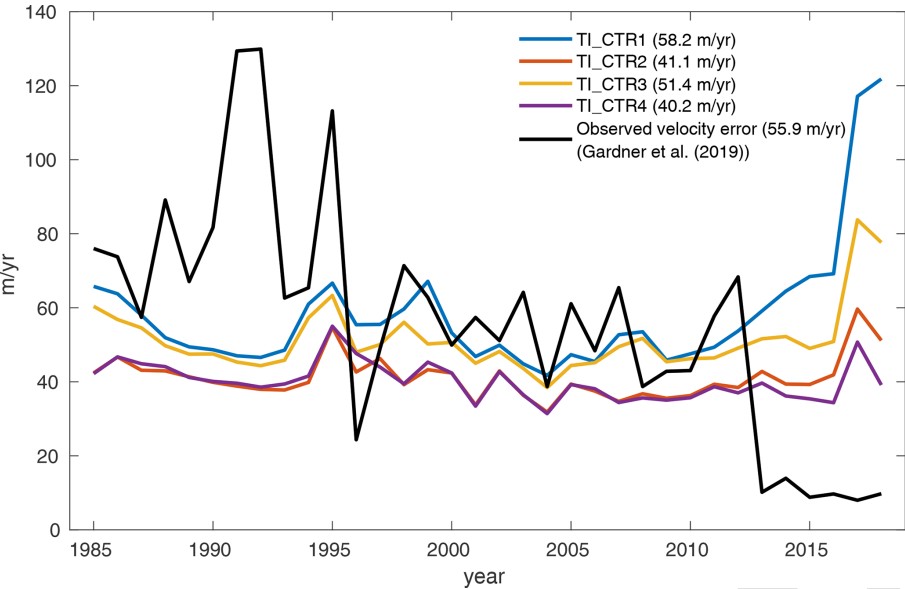

**Figure 8.** Time series of root mean square error between observed and modeled ice velocity from 1985 to 2018 for different combinations of static and transient controls (TI_CTR1–TI_CTR4). The mean error for each experiment is included in the legend. The black line represents the mean annual error (Gardner et al., 2019) of observed velocities for the basin.

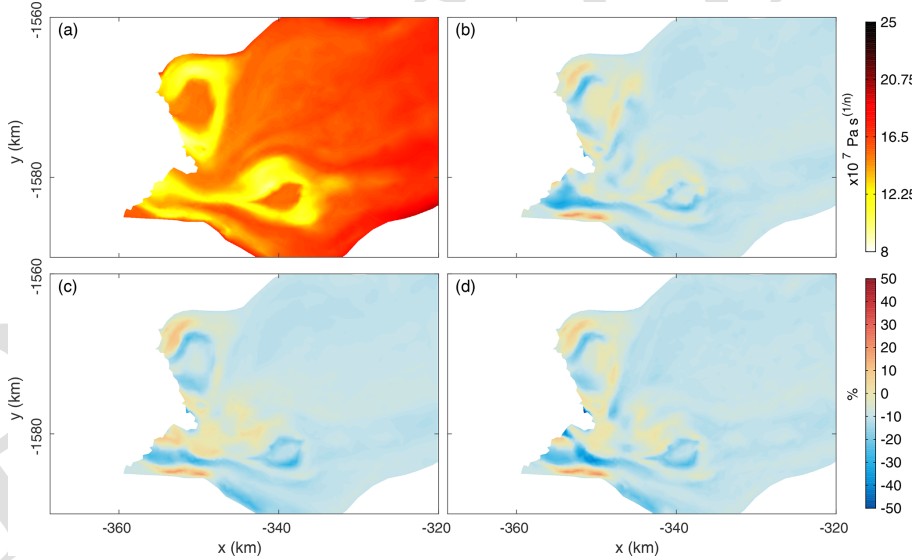

**Figure 9. (a)** Depth-averaged viscosity parameter, *B*, in 2018 from the thermal model used as the initial guess in the inversions. **(b, c, d)** Difference in the 2018 viscosity parameter compared to thermal model for transient-inversion-based simulations: **(b)** static friction coefficient and rheology parameter experiment (TI_CTR1), **(c)** static friction coefficient and transient rheology parameter experiment (TI_CTR3), and **(d)** transient friction coefficient and rheology parameter experiment (TI_CTR4).

vations and provide more confidence in near-future projections than the snapshot-calibrated simulations.

Our TI_PD experiments show that all projections based on the transient inversion provide an improved agreement with observations relative to those based on a snapshot inversion, even when the calibration period is only a few years. Although there were limited changes in velocity during the

short calibration period, the model initialized with transient calibration can still capture a significant acceleration after the calibration period. These results demonstrate that the simulations based on the transient inversion can enhance our confidence in near-future projections, even with a limited period of observations and when these observations include limited variability to properly calibrate the model. With the con-

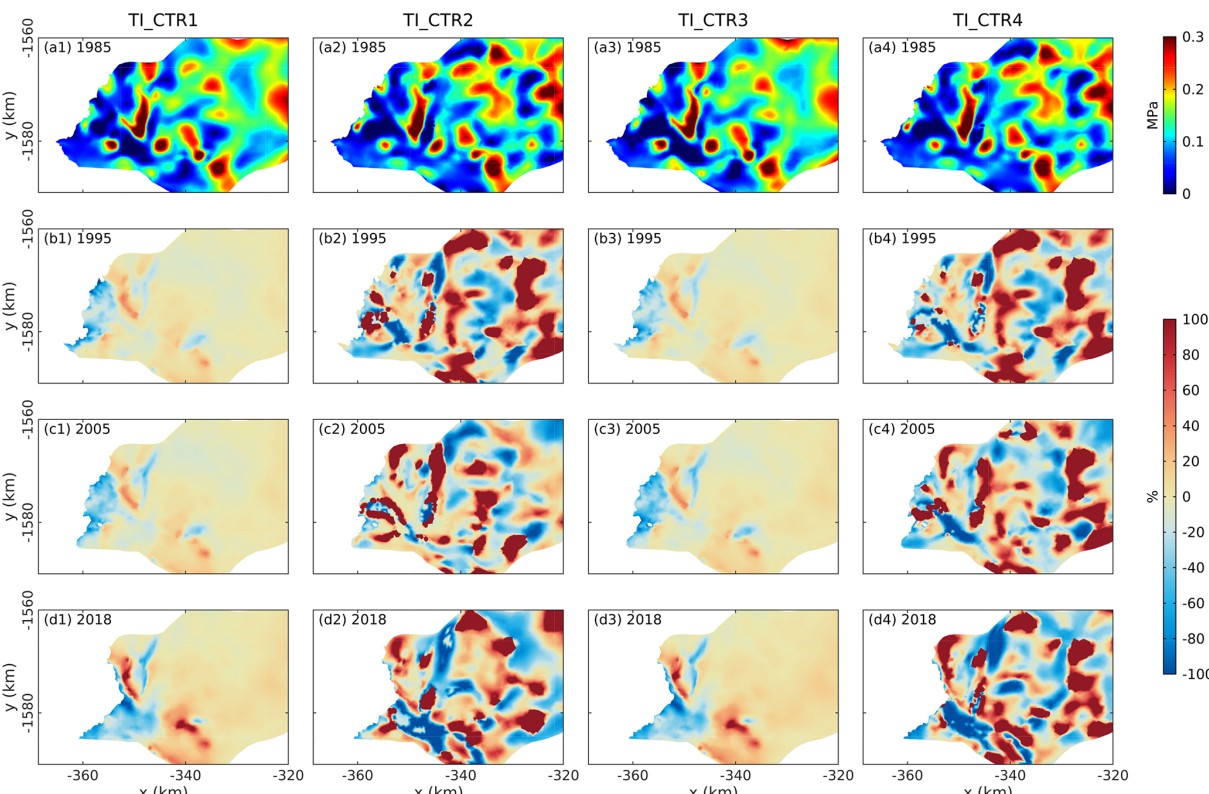

**Figure 10.** Evolution of basal stress between 1985 and 2018. First column: basal stress from the static friction coefficient and viscosity parameter experiment (TI_CTR1). **(a1)** Basal stress in 1985 and **(b1, c1, d1)** changes in basal stress, compared to 1985, in indicated years. Second column: **(a2)** basal stress in 1985 and **(b2, c2, d2)** its change for the transient friction coefficient and static viscosity parameter experiment (TI_CTR2). Third column: **(a3)** basal stress in 1985 and **(b3, c3, d3)** its change for the static friction coefficient and transient viscosity parameter experiment (TI_CTR3). Last column: **(a4)** basal stress in 1985 and **(b4, c4, d4)** its change for the transient friction coefficient and viscosity parameter experiment (TI_CTR4).

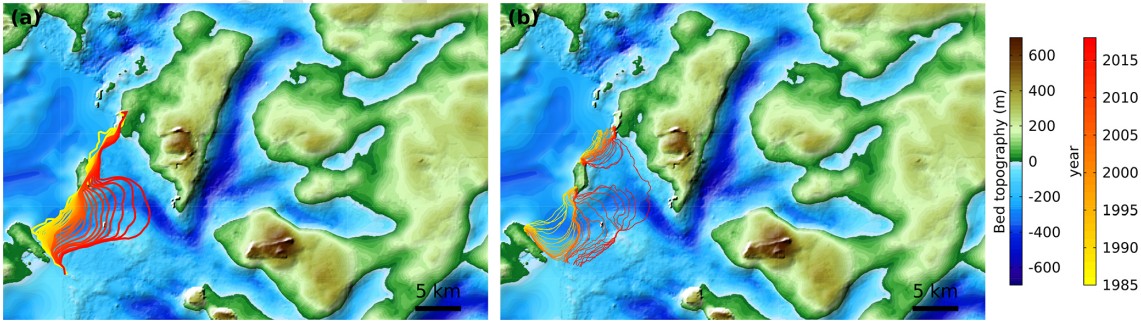

**Figure 11. (a)** Modeled ice front positions simulated from transient inversion for the calving law parameter (TI_Calving). **(b)** Observed ice front positions (Wood et al., 2021) from 1985 to 2018.

tinuous extension of observational records capturing recent changes in glaciers, the method presented here can be applied broadly to other glaciers to provide more reliable near-future projections.

AD is a lot more memory intensive than solving for the adjoint state explicitly or forward runs, which may limit the application of this approach to regional simulations or ice-sheet-wide-scale models with very coarse mesh resolution

(Morlighem et al., 2021). We additionally set up one simple experiment to investigate the scalability of the framework and the possibility to infer parameters on a coarse mesh and use them on a finer mesh. We interpolate the control parameter fields (i.e., $C$ and $B$) optimized with the transient inversion of our model (TI) above to a new finer-resolution mesh domain (100 m–5 km, $\sim 36\,000$ elements) and assess the new model performance to project future changes of the

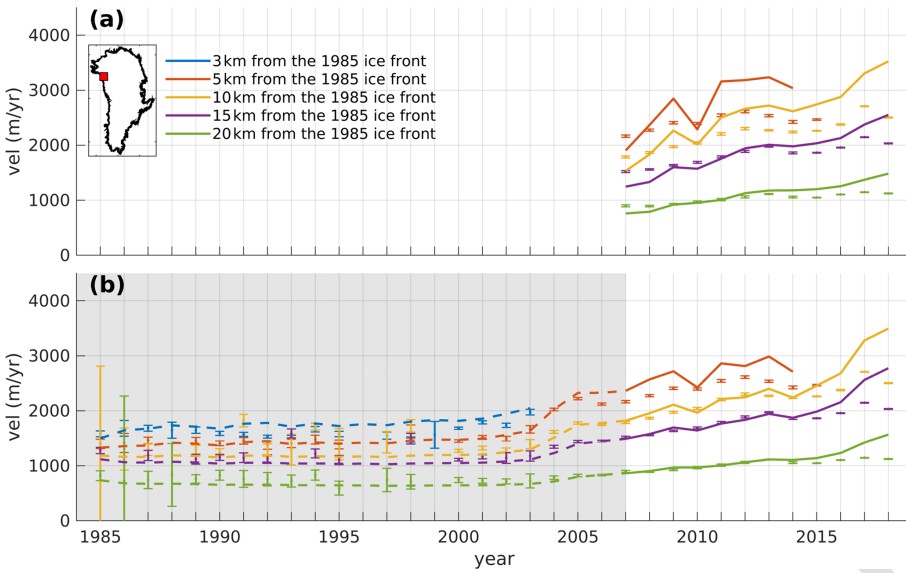

**Figure 12.** Same as Fig. 3 but for Sverdrup Glacier. The inset shows the location of Sverdrup Glacier and the gray box the period used for the transient inversion.

glacier (Fig. 13). The results show that the observed acceleration (94 %–105 % modeled-to-observed-velocity ratio in 2007) and mass loss from 2007 to 2018 (42 Gt) are better captured in the higher-resolution model with parameters interpolated from the coarse-resolution transient inversion than in the simulation based on the snapshot inversion (SI). These results suggest that the optimized parameters based on the coarse-resolution transient inversion could remain somewhat robust in a finer-resolution model, which is consistent with results from Barnes et al. (2021). Additional work on scalability of the framework should be conducted along with improvement of memory capability and code efficiency to be able to generalize AD to ice-sheet-wide simulations, but these results suggest a possible avenue to use AD in larger domains.

While transient inversions can potentially constrain time-varying, poorly known control parameters, a clear justification with physical constraints is needed. In this study, we allowed each parameter to vary every year arbitrarily. Our results from TI_CTR experiments show that allowing for time-varying control parameters only provides a small improvement of fit and causes the model to overfit the observations, which is consistent with results from Goldberg et al. (2015). This is a recurring problem in inverse modeling: a larger control space will lead to a better fit to the observations but at the expense of potentially unrealistic changes in control parameters. To avoid overfitting, more observations and physical interpretations are needed to better constrain temporal changes of these parameters. Additionally, further research is needed to identify the criteria to distinguish overfitting from observation-based uncertainties, although our model fit mostly falls within the range of the observation error.

In previous studies (e.g., Morlighem et al., 2016; Choi et al., 2018; Yu et al., 2019; Choi et al., 2021), calving laws have been calibrated by running transient simulations multiple times, manually adjusting the calving parameter and comparing to observed changes in ice front positions. In the framework presented in this study, however, optimizing the calving parameter can be done more efficiently, as the inversion automatically iterates to find the best fit to observations. For the TI_Calving experiment, we invert for the calving parameter after optimizing other control parameters, the friction coefficient and rheology parameter, rather than inverting for three control parameters simultaneously. We ran additional simulations (not shown here) to investigate the possibility for the simultaneous inversion for three control parameters using the sum of transient cost functions, $\mathcal{J}_{t3}(\boldsymbol{v}(t)) = \mathcal{J}_{t1}(\boldsymbol{v}(t)) + \mathcal{J}_{t2}(\boldsymbol{v}(t))$. The calibrated parameters depend strongly on choices of weights for each misfit term (e.g., $\gamma_1$, $\gamma_2$ and $\gamma_3$), which leads to several solutions for control parameters. Further research is required to better constrain these control parameters and investigate the best method to simultaneously infer a large number of parameters, as is done, e.g., for ocean state estimates (Forget et al., 2015).

## 5 Conclusions

In this study, we compare model initializations performed using snapshot and transient inversions to reproduce the recent changes of Kjer Glacier, West Greenland. We assess the impact of several cases for model initialization and future projections by conducting a suite of experiments using observational data from 1985 to 2018. These experiments show that

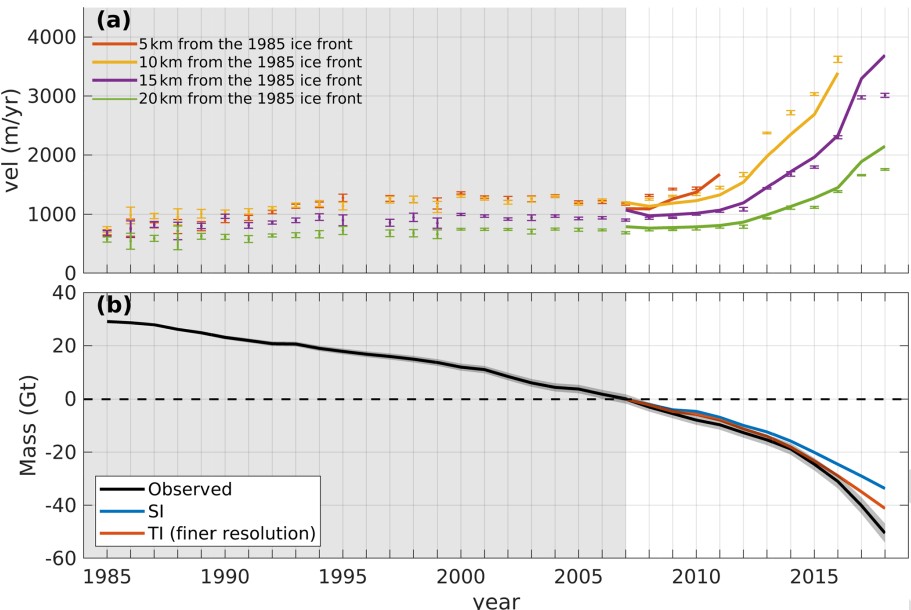

**Figure 13.** Comparison of model results with observations for the finer-resolution mesh domain of Kjer Glacier. **(a)** Comparison of observed (dots with error bars) and modeled (lines) velocities for the experiment utilizing control parameters from TI interpolated to the finer-resolution mesh. **(b)** Observed (black) and modeled (red) mass changes relative to 2007 for the same experiment. The blue line shows modeled mass change from SI shown in Fig. 4.

simulations based on transient inversions better capture the current trend of changes in glaciers and performed better for near-future projections, even when a short period of observations is used to constrain the simulation. Unlike the snapshot inversion that optimizes the model to a single moment in time, the transient inversion optimizes the model to fit multiple years of data, requiring the model to capture temporal variability in glacier flow. In the case of Kjer Glacier, this is achieved by softening the ice near the shear margins. This softening of the ice allows the model to better capture the glacier acceleration that occurs after the calibration period. Although large spatial and temporal variability in control parameters could improve the model fit to observations, it is essential to provide clear physical justification for temporally variable parameters to avoid overfitting. Additional experiments show that we can expand our initialization capabilities to infer calving parameters or use data assimilation on a coarse model and interpolate results onto a higher-resolution model. The methodology of transient inversion introduced in this study – which has not previously been applied to Greenland tidewater glaciers – could be applied to other regions of Greenland and to the ice-sheet-wide-scale model, which will take advantage of the wealth of remote sensing data that is currently available and will be available in the future.

## Appendix A: Regularization of inversion

We choose the regularization parameters in the inversion based on L-curve analysis (Hansen, 2001). For the snapshot

inversion, we first plot the L curve without the $B$ regularization term and choose the parameter for $C$ ($\gamma_3$ in Eq. 5). Once the optimal value for $\gamma_3$ is selected, we plot the L curve with the fixed $\gamma_3$ and choose the regularization parameter for $B$ ($\gamma_4$ in Eq. 5). The L curves used to choose the parameters are shown in Fig. A1.

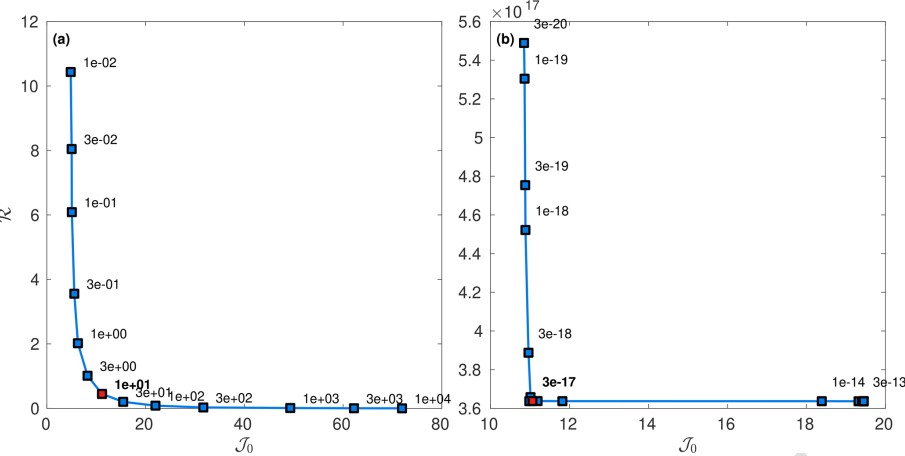

**Figure A1.** L curves for the snapshot inversion. **(a)** L curve for $\gamma_3$ and **(b)** L curve for $\gamma_4$. Points are labeled with values of $\gamma_3$ and $\gamma_4$, and the selected values are in red.

*Code and data availability.* The ISSM is open source and is available at http://issm.jpl.nasa.gov (last access: 20 April 2023). The ISSM version for this study is 4.23, corresponding to the public SVN repository tag number 27919. The source code for this version is also available at https://doi.org/10.5281/zenodo.8436924 (ISSM Team, 2023). Data for the main paper results and figures are available at https://doi.org/10.5281/zenodo.8436908 (Choi et al., 2023). BedMachine Greenland is freely available at the National Snow and Ice Data Center (NSIDC) (https://doi.org/10.5067/2CIX82HUV88Y, Morlighem et al., 2017a, b). Geothermal heat flux data are available at https://doi.org/10.17592/001.2018022701 (Greve, 2018). RACMO SMB information can be accessed at https://www.projects.science. uu.nl/iceclimate/models/racmo-archive.php (last access: TS3). Ice fronts data are available at https://doi.org/10.7280/D1667W (Wood et al., 2020). Annual ice velocity data and mass balance data are available at https://its-live.jpl.nasa.gov (Gardner et al., 2019) and https://doi.org/10.1073/pnas.1904142116 (Mouginot et al., 2019), respectively.

*Author contributions.* YC designed the experiments and conducted the modeling simulations with help from HS and NJS. MM contributed to the model development. AG provided the observational data. YC wrote the first version of the manuscript with inputs from HS, and all authors provided comments and edits to the manuscript.

*Competing interests.* The contact author has declared that none of the authors has any competing interests.

ther geographical representation in this paper. While Copernicus Publications makes every effort to include appropriate place names, the final responsibility lies with the authors.

*Acknowledgements.* This research was carried out at the Jet Propulsion Laboratory, California Institute of Technology, under a contract with the National Aeronautics and Space Administration. We acknowledge computational resources and support from the NASA Advanced Supercomputing Division.

*Financial support.* This research has been supported by the Jet Propulsion Laboratory's Greenland 2050 strategic Research and Technology Development program. Helene Seroussi was supported by a grant from the National Science Foundation's Navigating the New Arctic program (grant no. 2127246).

*Review statement.* This paper was edited by Jan De Rydt and reviewed by two anonymous referees.

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

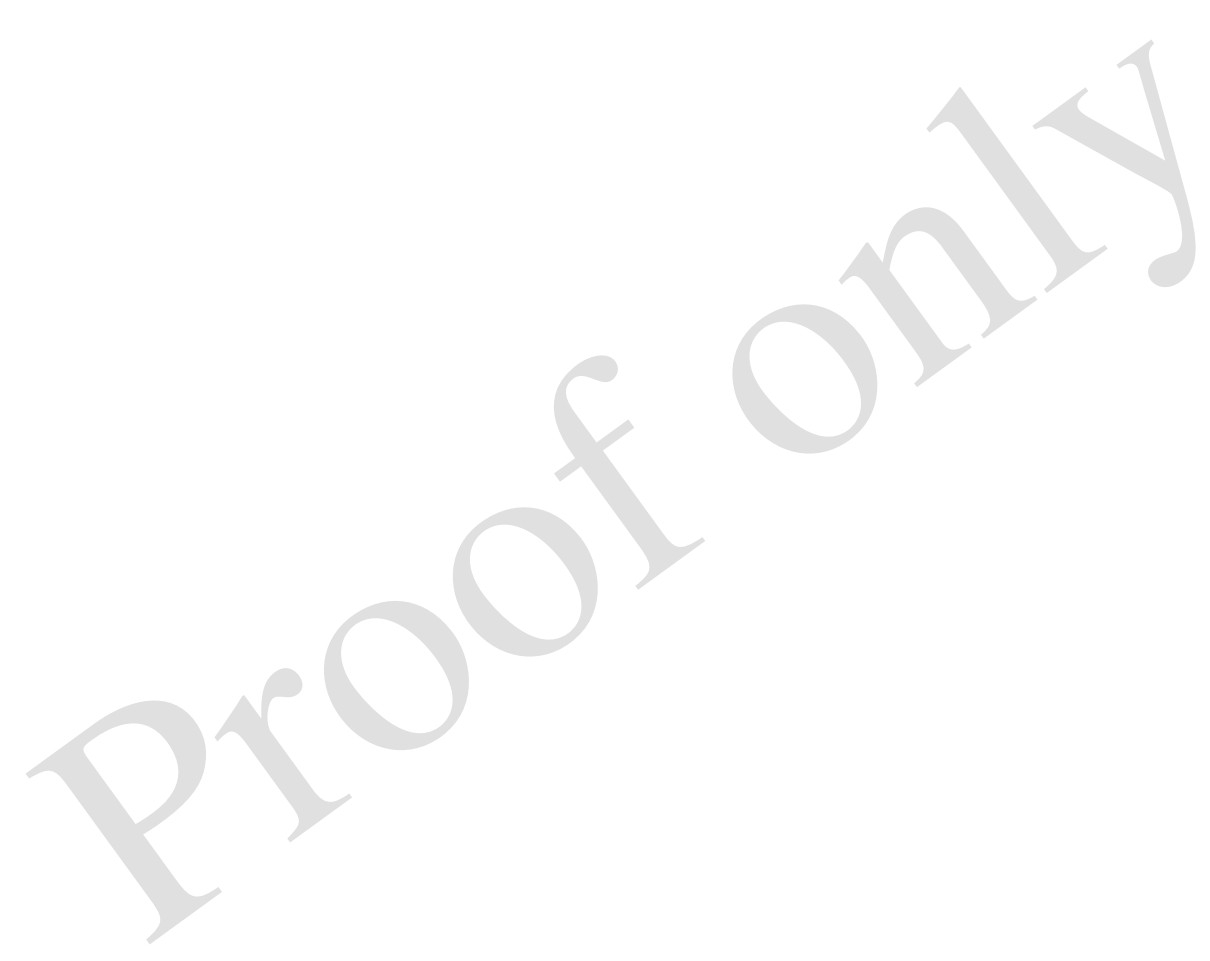

**Remarks from the typesetter**

TS1    Please give an explanation of why the equation needs to be changed. We have to ask the handling editor for approval. Thanks.

TS2    Please give an explanation of why the equation needs to be changed. We have to ask the handling editor for approval. Thanks.

TS3    Please provide date of last access.