# Peer review of "Impact of time-dependent data assimilation on ice flow model initialization and projections: A case study of Kjer Glacier, Greenland"

_The Cryosphere, 2023_

## Author Comment (AC1)

**Impact of time-dependent data assimilation on ice flow model initialization: A case study of Kjer Glacier, Greenland**
**– Authors' response (RC1) –**

Youngmin CHOI et al.

August 10, 2023

*This paper presents results from a transient inversion scheme which utilises automatic differentiation to initialise an ice flow model using multiple years of observed velocity data. Analysis is carried out to determine the effects of different lengths of observed data records, and whether the control variables are constant or varying in time. Comparisons are made with the more commonly used "snapshot" inversion, using only a single observational year. The conclusion is that the transient inversion method produces better results for capturing current trends and simulating the evolution of future ice flow, even with a fairly short observational record.*

*The manuscript is well written, and the premise of this study is very interesting. There are some nice results presented comparing the different approaches to transient inversion, and figures which display the information clearly. The subject matter is an important topic, and certainly within the scope of The Cryosphere.*

We thank the reviewers for reviewing this manuscript and their constructive comments.

*However, there is one major issue which I feel must be addressed. A notable difference between the snapshot and transient inversions in this study is that the snapshot inversion only inverts for C, keeping the value of B acquired from an estimate based on temperature. Meanwhile, the transient inversions invert for both B and C. I did not find any justification for this choice, which I imagine could be quite important. Without comparing a snapshot inversion which also inverts for both B and C, it appears to me that the comparison of methods is not like-for-like. Some proportion of the difference could (and I would have thought must) be due to the different treatment of B. It is noted by the authors in their discussion that some parts of the shear margins have a 45% reduction in the value of B after the transient inversions, which use the temperature-based estimate as an initial value. Unless I've missed something, from the information given in the current version of the paper,*

*there is no reason to think that a similar difference wouldn't occur when using a snapshot inversion
if the value of B was also inverted for in that case.*

*For me to find the results to convincingly support the conclusion in regard to snapshot vs. transient
inversion I would like to see the snapshot inversion performed inverting for both B and C, and then
one of the following as appropriate:*

1. *Results from the new snapshot inversion compared with the existing one to demonstrate that
   inverting for B causes negligible difference.*

2. *The result from the new snapshot used in the comparisons against the transient inversion
   results.*

*That being said, I do not contest that the transient inversion method does a good job, or that it
will likely still do better than a snapshot also inverting for B. I like the overall presentation of this
study, and believe other conclusions regarding the different approaches to transient inversion are
well supported. I was interested to see an inversion approach to calving parameters also, which is
an interesting addition to the study. I find very few issues with the rest of the manuscript and would
like to see it published, but have to recommend revision first to address my major issue above.*

We agree with the reviewer. We will run the new snapshot experiment that includes an inversion
for both $B$ and $C$, and add new results for that.

*Specific comments*

*Line 55 – "ice sheet models"*

We will change this in the revised text, as suggested.

*Figure 1 – It would be helpful to include the white line (2007 ice front, I assume, though this
should be clarified in the caption) underneath the coloured ice fronts in panel (b) for easy reference
between the two panels.*

We will add this to the revised text, as suggested.

*Line 92 – BedMachine citation appears to be in the wrong format.*

We will fix this in the revised text.

*Line 153 – Could the equation or chosen value for R be shown?*

We will add the equation for R, as suggested.

*Line 154 – For completeness, it would be good to show the L-curves and chosen values of $\gamma$ in an appendix/supplement.*

We will add the L-curves to the revised text, as also suggested by the reviewer 2.

*Line 179/Table 1 – Why is B not a control variable for the snapshot inversion? It is included in Eq.5, and inverted for in all other experiments. It's not clear to me why the temperature-based estimate is not used as an initial value as it is for the transient inversions. This relates to my major issue with the manuscript, detailed above.*

We will add this to the revised text as suggested.

*Figure 3 – While it is well explained in the caption, I wonder if a visual key/explanation could be added in some of the empty space of panel (a) to make it clear at a glance what the colours represent. Same for similar figures later on.*

We will add the legend for colors to the panel (a).

*Line 210 – "there still remain"*

We will fix this in the revised text, as suggested.

*Line 274 – I don't think "compared to the northern branch" is needed here, since it is immediately discussed in the next sentence. And comparing it to a low bar could detract from the point that the result for that area is quite good.*

We will fix this sentence in the revised text to make it clear.

*Figure 11 – Could this be displayed side by side with observed ice fronts for easy comparison? It would avoid having to scroll back up to Fig. 1!*

We will add the observed ice front positions next to the modeled ice front positions.

*Line 344-6 – The point about softening of the shear margins again draws my attention to the fact that B was not treated in the same way in snapshot and transient inversions. Perhaps the shear margins would have been softened to some extent in a snapshot inversion for B?*

80  We will explain this along with the revisions for new comparison between snapshot and transient

81  inversions above.

---

## Author Comment (AC2)

**Impact of time-dependent data assimilation on ice flow model initialization: A case study of Kjer Glacier, Greenland**
**– Authors' response (RC2) –**

Youngmin CHOI et al.

August 10, 2023

*In this study, authors make use of the vast amount of spatial and temporal coverage of satellite ice velocity observations and ice front positions of the Kjer Glacier (West Greenland). With the goal of improving the glacier's initial state and projections using transient inversions of the control parameters (the ice viscosity parameter B and the friction parameter C) in the model. The authors show that their methods can be applicable to two glaciers in the region. They also explore the possibility of including the stress threshold ($\sigma_{max}$) of the calving law as an additional control parameter while using the static friction coefficient (C) and viscosity parameter (B) obtained from the transient inversions (T1 in Table 1). Finally, the authors explore the possibility of inverting for all control parameters at once (C, B, and $\sigma_{max}$).*

*They conclude that transient inversions (on B and C) are able to capture the current trend of changes in glacier velocity better than snapshot inversions, and that those transient inversions improve the models ability to predict near-future changes. Even if a short period of observations is used for the calibration.*

*An additional experiment on the calving control parameter ($\sigma_{max}$) shows that it is possible to invert for this poorly constrained parameter via data assimilation techniques and reproduce to a certain extent the retreat of the Kjer glacier.*

*They also imply in their conclusion (this is not clearly stated) that the calibrated parameters depend strongly on the strength of the regularisation imposed (choice of weights) for each misfit term in the Cost functions, which leads to several solutions for control parameters and to an overfitting, if L-curve analysis is used to estimate the strength of the regularisation.*

*Overall, I find the manuscript well written, with a clear narrative and description of the methods and experiments. I also find the whole manuscript very interesting to read. I learned a lot!*

*I will definitely recommend the publication of the manuscript after the authors clarify some of my*
*questions below and make some minor changes.*

We thank the reviewers for reviewing this manuscript and thier constructive comments.

*Main comment:*

*The authors do not describe how the L-curve criteria has been applied in their study. I think this*
*should be explained in Section 2.4 (L151-162). There is no information on the values of the ($\gamma$) and*
*no L-curves are shown. There should be some information on how these parameters are chosen.*
*In other words, how the authors choose the strength of their regularisation in each Cost function?*
*Maybe some explanation similar to previous studies that use L-curve analysis (Gillet-Chaulet et al.*
*2012; Seddik et al. 2017; Barnes et al. 2021).*

*Probably authors could also add a table in the annex with the $\gamma$ parameter values and the L-curves*
*(or L-surface if that is the case) and describe what criteria they used for choosing $\gamma$ values and*
*if they keep the same values for all the experiments. They mention some overfitting and that more*
*investigation is needed in this area, I think this is an important point and should be highlighted.*

We agree with the reviewer regarding this point. This is also suggested by the other reviewer. We
will add the L-curve plot figure and explain how $\gamma$ was chosen. We kept the same values for this
study and we will add that to the revised text as well.

*Is also not clear to me why in the SI experiment, the authors do not invert for the ice viscosity*
*parameter (B) and estimate B from modelled ice temperature instead (and only in that experiment).*
*This will just add extra uncertainties to the inverted field (i.e. errors in the ice temperature model*
*will be propagated to the results). This error could be difficult to account for and might influence*
*the results shown in Figure 3 for the SI inversion. Clarifying that will strengthen the results of the*
*manuscript.*

We agree with the reviewer. We will run the new snapshot simulation that includes the inversion
for the ice viscosity parameter $(B)$, and add those results.

*Title suggestion: maybe this should be initialization and projections (or forecast).*

We will change the current title to "Impact of time-dependent data assimilation on ice flow model
initialization and projections: A case study of Kjer Glacier, Greenland", as suggested.

*L17: "accurate mass balance" $->$ "accurate ice sheet mass loss"*

We will change this in the revised text, as suggested.

*L30: "but often fail at accurately capturing their present-day configuration", add citation.*

We will add it to the revised text.

*L45-L60: literature review, probably I missed this but it could be nice if the authors relate those studies to transient inversions (what studies use that type of calibration technique, additionally to the use of AD and data assimilation).*

We will clarify this in the revised text.

*L130: Remind the reader what parameters you are inverting for? It will be good to mention this also in the Introduction.*

We will add it to the revised text.

*L144-146: "This approach allows to better understand the physical process involved in reproducing the ice stream..." Point to evidence of this in the results section.*

We will add it to the revised text, as suggested.

*L190: "limit uncertainties from calving parametrisations", I will add (this is optional): that it also avoids having to reconcile the SMB (estimated by RACMO) with the mass loss estimated by the calving law.*

We will add it to the revised text, as suggested.

*L283-284: "which improves the model's ability" − > "which improves confidence in the model's ability to provide realistic near-future projections". Maybe mention that calibration error and its influence on the model projections still needs to be quantified.*

We will change this in the revised text, as suggested. We will also mention the calibration error and its influence.

*L289: "... 2007 to 2018 is overestimated" indicate the colour of the line in the figure.*

We will add this to the revised text.

*L299-L301: "These results demonstrate that the simulations based on the transient inversion can enhance our confidence in near-future projections, even with a limited period of observations and when these observations include limited variability to properly calibrate the model".*

*What happens if the observations used for the transient inversions have a lot of variability in ice velocity? For example if you were to use 2010-2013 (where there is more variability than the periods used for Fig 5) would the model be able to predict changes in the following years?*

We expect the model is able to predict changes after the inversion period. To show this, we will run additional experiments and add those results.

*L306: It will be nice to add a comment (though this is optional as it is not the goal of the study) regarding the quantification of calibration uncertainty in transient inversions and the propagation of this type of error to projections. The error in the inverted parameters for this type of calibration will be very expensive to quantify via state-of-the-art Markov chain Monte Carlo (MCMC) methods (Tierney, 1994. Petra et al. 2014) and/or Hessian-based Bayesian approaches (Isaac et al., 2015, Koziol et al., 2021), as they will require multiple evaluations of the forward model to sample all the variability in the parameter space. For snapshot inversions the forward model is just a single velocity solved and for transient inversions this forward model is a sequence of time steps. Thus very expensive for error quantification in large-scale inverse problems (¿ 100, 000 mesh elements). Probably this is a limitation for large scale ice sheet problems but might be possible for marine-terminating glaciers elsewhere.*

This is an interesting point and we will add a comment about uncertainty quantification to the revised text.

*L346: The authors write: "Although large spatial and temporal variability in control parameters could improve the model fit to observations, clear physical justification should be made to avoid overfitting". "Physical justification" of what? I get a bit lost in this statement.*

We meant the "physical justification of changing control parameters every year" as we did in TR_CTR experiments. We will clarfy this in the revised manuscript.

*Figures*

*Figure 3, 5, 7, 12 and 13a, will benefit by including in the plots the uncertainty in the ITS_LIVE dataset (ideally the standard deviation of the data set) this could be added to the plot by either using error bars in a scatter plot or changing the size of the triangles according to the error in the data base? This will help us identify if model results are within the observations uncertainty at a given location (and time).*

We will add this to the revised manuscript.

*Figure 4, 6 and 8. Add citation to the legend for the observations.*

We will add it to the revised manuscript, as suggested.

*Figure 10. There is a mistake in the caption for the third column, seems like it has the same*
*as the Second column caption but they are different experiments according to Table 1. Check for*
*inconsistencies with Table 1.*

We will fix this in the revised manuscript.

---

## Author Response (AR1)

**Impact of time-dependent data assimilation on ice flow model initialization: A case study of Kjer Glacier, Greenland – Authors' response –**

Youngmin CHOI et al.

October 12, 2023

**1 Reviewer 1**

*This paper presents results from a transient inversion scheme which utilises automatic differentiation to initialise an ice flow model using multiple years of observed velocity data. Analysis is carried out to determine the effects of different lengths of observed data records, and whether the control variables are constant or varying in time. Comparisons are made with the more commonly used "snapshot" inversion, using only a single observational year. The conclusion is that the transient inversion method produces better results for capturing current trends and simulating the evolution of future ice flow, even with a fairly short observational record.*

*The manuscript is well written, and the premise of this study is very interesting. There are some nice results presented comparing the different approaches to transient inversion, and figures which display the information clearly. The subject matter is an important topic, and certainly within the scope of The Cryosphere.*

We thank the reviewer for reviewing this manuscript.

*However, there is one major issue which I feel must be addressed. A notable difference between the snapshot and transient inversions in this study is that the snapshot inversion only inverts for C, keeping the value of B acquired from an estimate based on temperature. Meanwhile, the transient inversions invert for both B and C. I did not find any justification for this choice, which I imagine could be quite important. Without comparing a snapshot inversion which also inverts for both B and C, it appears to me that the comparison of methods is not like-for-like. Some proportion of the difference could (and I would have thought must) be due to the different treatment of B. It is noted*

*by the authors in their discussion that some parts of the shear margins have a 45% reduction in the value of B after the transient inversions, which use the temperature-based estimate as an initial value. Unless I've missed something, from the information given in the current version of the paper, there is no reason to think that a similar difference wouldn't occur when using a snapshot inversion if the value of B was also inverted for in that case.*

*For me to find the results to convincingly support the conclusion in regard to snapshot vs. transient inversion I would like to see the snapshot inversion performed inverting for both B and C, and then one of the following as appropriate:*

1. *Results from the new snapshot inversion compared with the existing one to demonstrate that inverting for B causes negligible difference.*

2. *The result from the new snapshot used in the comparisons against the transient inversion results.*

*That being said, I do not contest that the transient inversion method does a good job, or that it will likely still do better than a snapshot also inverting for B. I like the overall presentation of this study, and believe other conclusions regarding the different approaches to transient inversion are well supported. I was interested to see an inversion approach to calving parameters also, which is an interesting addition to the study. I find very few issues with the rest of the manuscript and would like to see it published, but have to recommend revision first to address my major issue above.*

We agree with the reviewer. We conducted a new snapshot experiment that includes inversions for both $B$ and $C$. We updated the snapshot experiment with these new results, including in the figures and in the text. The new snapshot experiment introduces some changes to the patterns inferred and glacier evolution, but these changes are limited and do not impact the overall results.

*Specific comments*

*Line 55 – "ice sheet models"*

Done.

*Figure 1 – It would be helpful to include the white line (2007 ice front, I assume, though this should be clarified in the caption) underneath the coloured ice fronts in panel (b) for easy reference between the two panels.*

We added this to the revised figure caption, as suggested.

*Line 92 – BedMachine citation appears to be in the wrong format.*

We fixed this in the revised text.

*Line 153 – Could the equation or chosen value for R be shown?*

We added the equation instead of R, as suggested.

*Line 154 – For completeness, it would be good to show the L-curves and chosen values of $\gamma$ in an appendix/supplement.*

We added the L-curve analysis in the appendix.

*Line 179/Table 1 – Why is B not a control variable for the snapshot inversion? It is included in Eq.5, and inverted for in all other experiments. It's not clear to me why the temperature-based estimate is not used as an initial value as it is for the transient inversions. This relates to my major issue with the manuscript, detailed above.*

We changed the experiment to have B also inferred in the snapshot inversion and updated the table as well as the text, as suggested.

*Figure 3 – While it is well explained in the caption, I wonder if a visual key/explanation could be added in some of the empty space of panel (a) to make it clear at a glance what the colours represent. Same for similar figures later on.*

We added the legend for colors to Figure 3(a) and other similar figures (Fig. 5, Fig. 7, Fig. 12, and Fig. 13).

We fixed this in the revised text, as suggested.

*Line 274 – I don't think "compared to the northern branch" is needed here, since it is immediately discussed in the next sentence. And comparing it to a low bar could detract from the point that the result for that area is quite good.*

We fixed this sentence in the revised text to make it more clear.

*Figure 11 – Could this be displayed side by side with observed ice fronts for easy comparison? It would avoid having to scroll back up to Fig. 1!*

We added the observed ice front positions next to the modeled ice front positions.

*Line 344-6 – The point about softening of the shear margins again draws my attention to the fact that B was not treated in the same way in snapshot and transient inversions. Perhaps the shear margins would have been softened to some extent in a snapshot inversion for B?*

The new value of B in the new snapshot inversion decreased by up to 10% along the margins compared to the initial value. We updated this in the revised text.

**2   Reviewer 2**

*In this study, authors make use of the vast amount of spatial and temporal coverage of satellite ice velocity observations and ice front positions of the Kjer Glacier (West Greenland). With the goal of improving the glacier's initial state and projections using transient inversions of the control parameters (the ice viscosity parameter B and the friction parameter C) in the model. The authors show that their methods can be applicable to two glaciers in the region. They also explore the possibility of including the stress threshold ($\sigma_{max}$) of the calving law as an additional control parameter while using the static friction coefficient (C) and viscosity parameter (B) obtained from the transient inversions (T1 in Table 1). Finally, the authors explore the possibility of inverting for all control parameters at once (C, B, and $\sigma_{max}$).*

*They conclude that transient inversions (on B and C) are able to capture the current trend of changes in glacier velocity better than snapshot inversions, and that those transient inversions improve the models ability to predict near-future changes. Even if a short period of observations is used for the calibration.*

*An additional experiment on the calving control parameter ($\sigma_{max}$) shows that it is possible to invert for this poorly constrained parameter via data assimilation techniques and reproduce to a certain extent the retreat of the Kjer glacier.*

*They also imply in their conclusion (this is not clearly stated) that the calibrated parameters depend strongly on the strength of the regularisation imposed (choice of weights) for each misfit term in the Cost functions, which leads to several solutions for control parameters and to an overfitting, if L-curve analysis is used to estimate the strength of the regularisation.*

*Overall, I find the manuscript well written, with a clear narrative and description of the methods and experiments. I also find the whole manuscript very interesting to read. I learned a lot!*

*I will definitely recommend the publication of the manuscript after the authors clarify some of my questions below and make some minor changes.*

We thank the reviewer for reviewing this manuscript and for the constructive comments.

*Main comment:*

*The authors do not describe how the L-curve criteria has been applied in their study. I think this should be explained in Section 2.4 (L151-162). There is no information on the values of the ($\gamma$) and no L-curves are shown. There should be some information on how these parameters are chosen. In other words, how the authors choose the strength of their regularisation in each Cost function? Maybe some explanation similar to previous studies that use L-curve analysis (Gillet-Chaulet et al. 2012; Seddik et al. 2017; Barnes et al. 2021).*

*Probably authors could also add a table in the annex with the $\gamma$ parameter values and the L-curves (or L-surface if that is the case) and describe what criteria they used for choosing $\gamma$ values and if they keep the same values for all the experiments. They mention some overfitting and that more investigation is needed in this area, I think this is an important point and should be highlighted.*

We agreed with the reviewer regarding this point. This was also suggested by the other reviewer. We added the L-curve analysis in the appendix and explained how the $\gamma$ coefficients were chosen. We kept the same values for this study and we added that to the revised text as well.

*Is also not clear to me why in the SI experiment, the authors do not invert for the ice viscosity parameter (B) and estimate B from modelled ice temperature instead (and only in that experiment). This will just add extra uncertainties to the inverted field (i.e. errors in the ice temperature model will be propagated to the results). This error could be difficult to account for and might influence the results shown in Figure 3 for the SI inversion. Clarifying that will strengthen the results of the manuscript.*

We ran a new snapshot simulation that includes the inversion for the ice viscosity parameter ($B$), and updated the figures and text with these new results; these changes only have a limited impact on the results and do not impact the conclusions of our study.

*Title suggestion: maybe this should be initialization and projections (or forecast).*

This is a good idea, so we changed the current title to "Impact of time-dependent data assimilation on ice flow model initialization and projections: A case study of Kjer Glacier, Greenland", as suggested.

*L17: "accurate mass balance" $->$ "accurate ice sheet mass loss"*

We changed this in the revised text, as suggested.

*L30: "but often fail at accurately capturing their present-day configuration", add citation.*

We added it to the revised text.

*L45-L60: literature review, probably I missed this but it could be nice if the authors relate those studies to transient inversions (what studies use that type of calibration technique, additionally to the use of AD and data assimilation).*

We revised the text accordingly.

*L130: Remind the reader what parameters you are inverting for? It will be good to mention this also in the Introduction.*

We added it to the revised text.

*L144-146: "This approach allows to better understand the physical process involved in reproducing the ice stream..." Point to evidence of this in the results section.*

We added examples of physical processes that we want to calibrate in this study.

*L190: "limit uncertainties from calving parametrisations", I will add (this is optional): that it also avoids having to reconcile the SMB (estimated by RACMO) with the mass loss estimated by the calving law.*

We added it to the revised text, as suggested.

*L283-284: "which improves the model's ability" − > "which improves confidence in the model's ability to provide realistic near-future projections". Maybe mention that calibration error and its influence on the model projections still needs to be quantified.*

We changed this in the revised text, as suggested. We also mentioned the calibration error and its influence.

*L289: "...2007 to 2018 is overestimated" indicate the colour of the line in the figure.*

We added this to the revised text.

*L299-L301: "These results demonstrate that the simulations based on the transient inversion can enhance our confidence in near-future projections, even with a limited period of observations and when these observations include limited variability to properly calibrate the model".*

*What happens if the observations used for the transient inversions have a lot of variability in ice velocity? For example if you were to use 2010-2013 (where there is more variability than the periods used for Fig 5) would the model be able to predict changes in the following years?*

We added experiments that used 2010-2013 velocities to calibrate the model (TI_PD4). The model still effectively captures the acceleration after the inversion period but display more variability and increased acceleration. We added these results to the revised figure (Fig. 5) and text.

*L306: It will be nice to add a comment (though this is optional as it is not the goal of the study) regarding the quantification of calibration uncertainty in transient inversions and the propagation of this type of error to projections. The error in the inverted parameters for this type of calibration will be very expensive to quantify via state-of-the-art Markov chain Monte Carlo (MCMC) methods (Tierney, 1994. Petra et al. 2014) and/or Hessian-based Bayesian approaches (Isaac et al., 2015, Koziol et al., 2021), as they will require multiple evaluations of the forward model to sample all the variability in the parameter space. For snapshot inversions the forward model is just a single velocity solved and for transient inversions this forward model is a sequence of time steps. Thus very expensive for error quantification in large-scale inverse problems (¿ 100, 000 mesh elements). Probably this is a limitation for large scale ice sheet problems but might be possible for marine-terminating glaciers elsewhere.*

We added some discussion about uncertainty quantification for future research in the Discussion section.

*L346: The authors write: "Although large spatial and temporal variability in control parameters could improve the model fit to observations, clear physical justification should be made to avoid overfitting". "Physical justification" of what? I get a bit lost in this statement.*

We meant the "physical justification of changing control parameters every year" as we did in TR_CTR experiments. We revised the text.

*Figures*

*Figure 3, 5, 7, 12 and 13a, will benefit by including in the plots the uncertainty in the ITS_LIVE dataset (ideally the standard deviation of the data set) this could be added to the plot by either using error bars in a scatter plot or changing the size of the triangles according to the error in the data base? This will help us identify if model results are within the observations uncertainty at a given location (and time).*

198  We added the error bars in those figures.

199  *Figure 4, 6 and 8. Add citation to the legend for the observations.*

200  We added citations to the revised figures, as suggested.

201  *Figure 10. There is a mistake in the caption for the third column, seems like it has the same*
202  *as the Second column caption but they are different experiments according to Table 1. Check for*
203  *inconsistencies with Table 1.*

204  We fixed the table in the revised manuscript.